# Safe and Effective Cynomolgus Monkey GLP—Tox Study with Repetitive Intrathecal Application of a TGFBR2 Targeting LNA-Gapmer Antisense Oligonucleotide as Treatment Candidate for Neurodegenerative Disorders

**DOI:** 10.3390/pharmaceutics14010200

**Published:** 2022-01-15

**Authors:** Sebastian Peters, Eva Wirkert, Sabrina Kuespert, Rosmarie Heydn, Siw Johannesen, Anita Friedrich, Susanne Mailänder, Sven Korte, Lars Mecklenburg, Ludwig Aigner, Tim-Henrik Bruun, Ulrich Bogdahn

**Affiliations:** 1Department of Neurology, University Hospital Regensburg, 93053 Regensburg, Germany; sebastian.peters@velvio.com (S.P.); Eva.wirkert@velvio.com (E.W.); sabrina.kuespert@velvio.com (S.K.); rosmarie.heydn@velvio.com (R.H.); siw.johannesen@bgu-murnau.de (S.J.); tim-henrik.bruun@velvio.com (T.-H.B.); 2Velvio GmbH, Am Biopark 11, 93053 Regensburg, Germany; 3BG Trauma Center, Professor Küntscher Str. 8, 82418 Murnau am Staffelsee, Germany; 4Granzer Regulatory Consulting & Services, Kistlerhofstr. 172C, 81379 München, Germany; Friedrich@granzer.biz (A.F.); Mailaender@granzer.biz (S.M.); 5Labcorp Early Development Services GmbH, 48163 Münster, Germany; sven.korte@labcorp.com (S.K.); lars.mecklenburg@labcorp.com (L.M.); 6Institute of Molecular Regenerative Medicine, Spinal Cord Injury and Tissue Regeneration Center Salzburg (SCI-TReCS), Paracelsus Medical University Salzburg, 5020 Salzburg, Austria; ludwig.aigner@pmu.ac.at

**Keywords:** TGFβ-signaling, neurogenesis, neurodegeneration, amyotrophic lateral sclerosis, GLP-ToxStudy, antisense oligonucleotide therapy, LNA

## Abstract

The capability of the adult central nervous system to self-repair/regenerate was demonstrated repeatedly throughout the last decades but remains in debate. Reduced neurogenic niche activity paralleled by a profound neuronal loss represents fundamental hallmarks in the disease course of neurodegenerative disorders. We and others have demonstrated the endogenous TGFβ system to represent a potential pathogenic participant in disease progression, of amyotrophic lateral sclerosis (ALS) in particular, by generating and promoting a disequilibrium of neurodegenerative and neuroregenerative processes. The novel human/primate specific LNA Gapmer Antisense Oligonucleotide “NVP-13”, targeting TGFBR2, effectively reduced its expression and lowered TGFβ signal transduction in vitro and in vivo, paralleled by boosting neurogenic niche activity in human neuronal progenitor cells and nonhuman primate central nervous system. Here, we investigated NVP-13 in vivo pharmacology, safety, and tolerability following repeated intrathecal injections in nonhuman primate cynomolgus monkeys for 13 weeks in a GLP-toxicology study approach. NVP-13 was administered intrathecally with 1, 2, or 4 mg NVP-13/animal within 3 months on days 1, 15, 29, 43, 57, 71, and 85 in the initial 13 weeks. We were able to demonstrate an excellent local and systemic tolerability, and no adverse events in physiological, hematological, clinical chemistry, and microscopic findings in female and male Cynomolgus Monkeys. Under the conditions of this study, the no observed adverse effect level (NOAEL) is at least 4 mg/animal NVP-13.

## 1. Introduction

Disruptions and impairments of the central nervous system (CNS) are common phenomena of our society and the origin of a broad variety of somatic and affective pathologies including neurodegenerative disorders such as Alzheimer’s disease, Parkinson’s disease, Chorea Huntington, Multiple Sclerosis, and Amyotrophic Lateral Sclerosis (ALS). Two major common characteristics of all neurodegenerative disorders are (1) a disequilibrium of neuroregenerative and neurodegenerative processes with a shift towards the latter, and (2) a dysregulation of the transforming growth factor β (TGFβ) system [1,2,3,4,5,6]. ALS is a highly aggressive neurodegenerative disorder with an incidence of two to three patients per 100.000 population, a prevalence of four to eight per 100.000, and a lifetime risk of 1 in 350 to 450. The average time to diagnosis ranges from 8 to 15 months; patients of all ages may be affected (peak 58 to 63 years of age for sporadic disease and 47 to 52 years for familial disease) [7,8]. The median survival after diagnosis is 15.8 months, and approximately 10% of patients are long-term survivors [9]. In addition, there is a related family background with specific mutations in approximately 5–10% of cases [7]. However, more than 30 mutations are known to cause or to be associated with ALS [10]. The pathophysiology is complex and heterogeneous, but collectively, ALS leads to premature degeneration of motor neurons in the motor cortex and spinal cord (upper and lower motor neurons) in all patients [11,12]. Except for Riluzole (Rilutek) [13,14,15,16] and Edavarone [17], there is currently no drug approved in label for ALS.

Antisense oligonucleotide (ASO) technology, first introduced and described in 1978 by Zamecnik et al. in studies with Rous sarcoma virus [18,19], has consistently been developed over recent decades [20] with manifold clinical development activities [21,22,23,24,25]. The extreme specificity and selectivity for their targets, long half-lives in combination with low toxicity, and low adverse effects make these drugs candidates as highly promising therapeutics [23]. The most prominent contender so far is Nusinersen, a successful FDA-approved ASO and the first compound approved for treating spinal muscular atrophy [26,27,28,29,30]. At the moment, there is a tremendous gain in ASOs that have entered clinical trials for a broad variety of different disorders including CNS diseases, muscle disease, cardiovascular disease, metabolic diseases, eye diseases, tumor burden, infectious diseases, lung diseases, and immunological disorders [21,22,23], but also neurodegenerative disorders including Huntington’s disease [25] and ALS [24]. The latter ALS dedicated “Tofersen” specifically targets the most prominent ALS mutation SOD1_mt_—very recent preliminary clinical phase 3 data already reveal positive trends in clinical course and patient subscores (unpublished data). In contrast to ALS specific single mutation approaches, the NVP-13 concept presented here targets specifically a single signaling activity, thereby reaching multiple pathophysiological relevant downstream targets. In consequence, reactivated neurogenesis, reversal of fibrosis, immune modulation, and reactivation of autophagy are our prospective functional treatment goals.

The endogenous TGFβ system [31,32,33] is a powerful regulator of fundamental cellular and physiological processes including proliferation, cell differentiation, and growth. In addition, TGFβ is involved in immune regulation, stem cell activity, and fibrosis resp. extracellular matrix. As demonstrated from other molecular players, the effects are very context-, dose-, and time-dependent. Thus, depending on concentration and context, TGFβ effects may be either beneficial (neuroprotective, stem cell activating, anti-inflammatory, activating autophagy) or detrimental (neurodestructive, stem cell arresting, pro-inflammatory, pro-fibrotic, inhibitory for autophagy) [34,35,36]. Due to the chronic systemic and local neuro-inflammatory milieu, neurodegenerative disorders exhibit an upregulated TGFβ system. Consequently, a persistent dysregulation may result in a disturbed homeostasis on several levels, resulting in an imbalance of degenerative and regenerative processes [37]. We recently could show that, compared to healthy controls, TGFβ and its receptor TGFBR2 are upregulated in post mortem CNS tissue of ALS patients, with subsequent decreased stem cell activities in brain and SC [1]. A more recent review highlighted the TGFβ system to be altered and crucially involved in the ALS disease course, with dysfunctional signaling in early stages and a persistent over-activation at the clinical stage of disease [38].

As a consequence of the grave prognosis of ALS, a constantly enhanced TGFβ system activity in ALS patients, and the promising characteristics of ASO technology to downregulate critical targets, we developed a novel LNA-gapmer ASO targeting TGFBR2 (formerly TGF-βRII) mRNA (NVP-13). We previously showed that treatment with NVP-13 targeting the TGFBR2, reduced TGFBR2 expression, downstream receptor signaling, modulating intracellular fibrosis- and stem cell niche markers towards a more stem cell-favoring milieu in human neuronal precursor cells (ReNcell CX^®^ cells) in vitro [39]. In the same animals described in the present study, we were able to confirm our in vitro data in an in vivo approach, where we could specifically downregulate TGFBR2 mRNA and lower protein expression levels, leading to a dose-dependent upregulation of the neurogenic niche activity within hippocampus and subventricular zone of male and female cynomolgus monkeys [40]. The main goals of the current study were to determine the pharmacology, safety, and tolerability of repeated intrathecal NVP-13 administrations within a classical regulatory 13 weeks GLP-toxicology study design in order to support the first clinical trials in ALS patients.

## 2. Materials and Methods

### 2.1. Antisense Oligonucleotide (NVP-13) Characteristics

NVP-13, an LNA-gapmer Antisense Oligonucleotide (ASO), was designed as the drug candidate to specifically hybridize with the mRNA for human TGFBR2. The ASO exerts no cross-reactivity in non-primate species, including rodents, due to this high specificity. Therefore, non-human primates such as the cynomolgus monkeys must be used exclusively for the GLP-Tox program. The rationale for the selection of dose levels in this study was based on in vitro data (human neuronal progenitor cells) and on recently published data for the treatment of spinal muscular atrophy (SMA) [26,27,28,29,30] as well as on the experience of a screening study (Peters et al.: in preparation). The LNA ASO of this study has group chemical similarity with the 2-MOE ASO of the published SMA data, so that it could be referenced for the conceptualization of the GLP-program. The 5’and 3’wings of the ASO consist of locked nucleic acids to protect the ASO from degradation by exonucleases. LNA and deoxyribonucleic acid (DNA) are linked by a phosphorothioate backbone (PTO) [39]. In addition, the outcome of the screening study showed that all selected dose levels were expected to produce a pharmacodynamic effect, while the highest dose level achieves large exposure multiples, as stated in ICH guideline M3(R2). The route of administration was chosen because it is the intended human therapeutic route.

### 2.2. In Vivo Experimental Design

In vivo experimental design was performed as described previously [40]. Briefly, 0.9% NaCl (control group) and NVP-13 dissolved in 0.9% NaCl (three different doses) were injected repeatedly over a 13-weeks approach. Physical/neurological parameters (general sensomotor aspects, cerebral reflexes (pupillary, orbicularis oculi), and spinal reflexes (patellar, anal) incl. foot grip reflex, as well as abdominal palpation, body temperature, and heart and lung auscultation were investigated directly and 4 h after administration. If neurological abnormalities were present 4 h after dosing, additional neurological examination time points (in daily intervals) were performed to assess their reversibility/progression. For hematology, clinical chemistry, and CFS parameter investigation, blood and CSF samples were taken before every dosing, and tissue samples (liver, kidney, spinal cord, brain) were collected in the end of the study.

### 2.3. Pre-Study Procedures

Pre-study procedures were performed as described previously [40]. Briefly, for each GLP-Tox treatment group, selected male (n = 5) and female (n = 5) cynomolgus monkeys (Macaca fascicularis, supplied from a self-sustaining colony in Asia) aged 2 to 6 years, weighing 2.2 to 5 kg, were housed in groups of 3 animals. Animals were randomized to treatment groups to ensure equal bodyweight per group and sex. All animals were kept in an AAALAC accredited facility under standard laboratory conditions in a climate-controlled room with a minimum of 8 air changes/hour (12 h light/dark cycle, 19 to 25 °C, 40 to 70% humidity). All animals received a certified lab diet for primates (LabDiet 5048) twice daily, supplemented by fresh fruits and vegetables and had access to tap water ad libitum. Immediately after each handling/manipulation, the animals received a tasty reward. The non-human primate study was conducted following approval by an Institutional Animal Care and Use Committee and by the local authority (Landesamt für Natur, Umwelt und Verbraucherschutz) and was registered to the following file number: 84-02.04.2017.A093.

This study was performed in compliance with the following guidelines or recommendations concerning preclinical development of human pharmaceuticals:European Directive 2001/83/EC and all subsequent amendmentsGerman Drug LawInternational Conference on Harmonization (ICH) Guideline: Guidance on Non clinical Safety Studies for the Conduct of Human Clinical Trials and Marketing Authorization for Pharmaceuticals, M3(R2), issued in EMA as CPMP/ICH/286/95.ICH-S3A, Toxicokinetics: A Guidance for Assessing Systemic Exposure in Toxicology Studies, issued in EMA as CPMP/ICH/384/95ICH-S6, Preclinical Safety Evaluation of Biotechnology-Derived Pharmaceuticals, issued in EMA asCPMP/ICH/302/95, and first revision, issued in EMA as CHMP/ICH/731268/1998Guideline on repeated dose toxicity, issued by EMA as CPMP/SWP/1042/99 Rev 1

### 2.4. Dosing Procedure

The test item (1, 2, or 4 mg NVP-13/animal) or artificial cerebrospinal fluid—aCSF (in the control group) was administered intrathecally within 3 months on days 1, 15, 29, 43, 57, 71, and 85 in the initial 13 weeks. (Figure 1A). For intrathecal dosing, animals were anesthetized with ketamine and medetomidine and subsequently fixed an a specific injection chair. After disinfecting the skin, a micro incision of the skin was conducted using a 20 G needle. A pencil-point needle for pediatric use (Pencan Paed^®^ needle, 25 G, B. Braun Melsungen AG, Germany) was then inserted at the level of L4/L5, until CSF flow was observed (L3/4, L2/L3 were used if no CSF flow was observed). A volume of 0.5 mL CSF was removed for analytical purposes and to adjust for the volume addition by the test article. Thereafter, 0.75 mL of test item or aCSF (artificial cerebrospinal fluid; aCSF, from RD Systems Inc., Minneapolis, MN, USA) were injected over approximately 1 min, followed by a 0.25 mL flush with aCSF in all animals to empty the needle (Figure 1B). Needle and syringe were left in the dosing site for at least 30 s after the aCSF flush. Bepanthen^®^ aseptic wound ointment (contains chlorhexidine and dexpanthenol) was applied immediately after each dosing. Thereafter, the animal was placed in a lying position for 15 min before the antidote Atipamezole was applied.

### 2.5. Plasma Collection

Blood samples (1 mL for at least 0.4 mL plasma) were collected from the *vena cephalica antebrachii* or *vena saphena* into K_2_EDTA tubes from all animals at several time points. Blood samples were stored on crushed ice until further processing. Plasma was subsequently obtained by centrifugation at 4 °C and approximately 2300× *g* for 10 min. The plasma was aliquoted into four labeled micro tubes (three aliquots with at least 0.1 mL each and one aliquot with the remaining fluid) and stored frozen at −70 °C or below (Appendix A).

### 2.6. Cerebrospinal Fluid Collection

The CSF samples (approximately 0.3 mL; at the 24 h time point, only 0.25 mL) were collected by lumbar puncture from anesthetized and fixed animals into micro tubes (Sarstedt) at multiple time points (using the same spinal needle as for dosing). The CSF samples were frozen in liquid nitrogen and stored at −70 °C or below (Appendix A).

### 2.7. Tissue Collection

#### 2.7.1. Brain

At the time of euthanasia, the brain was sectioned in a brain matrix at 4 mm coronal slice thickness. Each coronal slice was further divided by a sagittal orientation into right and left hemispheres. The left hemisphere was preserved in 10% neutral buffered formalin for microscopic evaluation. The right hemisphere was frozen at −70 °C or below for bioanalytical, mRNA, and biomarker analysis.

#### 2.7.2. Spinal Cord

The spinal cord was divided into cervical, thoracic, and lumbar sections. Each section was further divided into two portions. A 3 cm portion from the rostral end of each section was frozen at −70 °C for bioanalytical, mRNA, and biomarker analysis, and the remaining portion was fixed in 10% neutral-buffered formalin for microscopic evaluation. In addition, a section for histology was taken proximal to the injection site (e.g., between L3 and L4).

#### 2.7.3. Liver and Kidney

The liver (left lobe) and the kidney (cortex and medulla) were collected. Samples of the liver and kidney were split into two aliquots of 0.25 g each; samples were frozen at −70 °C for bioanalytical analysis.

### 2.8. Clinical Observations

Throughout the entire study, all animals were observed twice a day for any signs of ill health or overt toxicity (clinical observations). In addition, animals were clinically examined in detail once a week. Feces were evaluated daily.

### 2.9. In Life Parameters

Standard in life parameters including body weight, food consumption, physical and neurologic examinations, neurobehavioral observations (Appendix A), ophthalmic observations, cardiovascular investigations (electrocardiography, blood pressure), and respiratory rate were performed on regular timepoints throughout the entire experiment.

### 2.10. Clinical Pathology

Blood samples (3.0 mL) were withdrawn from all animals once during the predose phase, during Week 13 of the dosing phase, and from all surviving animals during the last week of the recovery phase. Samples were collected from the *vena* cephalica antebrachii or vena saphena, and hematology tests, coagulation tests, and clinical chemistry tests were performed (Appendix A). The CSF samples were withdrawn from all animals once during the predose phase, in week 13 of the dosing phase, and from all surviving animals during the last week of the recovery phase. One aliquot of 0.5 mL was sampled for CSF chemistry and total cell counts (Appendix A). Withdrawn volume was substituted with aCSF. Urinary samples (after approximately 2 h without food) were collected from all animals once during the predose phase, during week 13 of the dosing phase, and from all surviving animals during the last week of the recovery phase (Appendix A).

### 2.11. End of in Life Phase

The in-life phase ended on the day following at least 13 weeks of dosing (Week 14, Day 92, 3 male and female animals of each group) or a 13-week recovery phase (Week 27, Day 183, remaining 2 male and female animals of each group). Necropsies were carried out in replicate order to ensure equal numbers of animals from each group and/or sex sacrificed on each day.

### 2.12. Necropsy, Organ Weights, and Macroscopic Observations

A full macroscopic examination was performed under the general supervision of a Pathologist.

### 2.13. Histology

Brain, spinal cord, and lymph node tissue from each animal were trimmed and embedded in paraffin wax, sectioned at a nominal 5 μm, stained with hematoxylin and eosin, and analyzed for microscopic alterations. Sectioning of the brain was performed such that, at a minimum, the following structures were captured: Neocortex (including frontal, parietal, temporal, and occipital cortex), paleocortex (olfactory bulbs and/or piriform lobe), basal ganglia (including caudate and putamen), limbic system (including hippocampus and cingulate gyri), thalamus/hypothalamus, midbrain regions including substantia nigra, subventricular zone, cerebellum, pons, and medulla oblongata.

### 2.14. Microscopic Observations

Tissues were examined microscopically by a pathologist.

### 2.15. NVP-13 Concentration in Plasma

The analytical work was performed in accordance with standard operating procedures, and for each chromatographic analysis batch, a detailed assay protocol was generated. The method was qualified previously at Axolabs under Method Development Project No. AN479.0-15 prior to the start of this study. The analytical method is based on an AEX-HPLC method with fluorescence detection that allows the sensitive and specific detection of NVP-13 from cynomolgus monkey plasma samples. The assay is based on the specific hybridization of a 16-mer complementary PNA-probe conjugated at the N-terminus with Atto425 dye. The duplex of PNA and parent compound yields a specific signal in the subsequent analysis by AEX-HPLC coupled to a fluorescence detector. Quantification is performed based on an external calibration curve generated from a standard dilution series in cynomolgus monkey plasma. Linear calibration curves (weighted 1/X) are calculated from 1 ng/mL to 5000 ng/mL.

### 2.16. Statistics

For graph design and statistical comparison, GraphPad Prism 8 was employed. All parameters were tested for Gaussian distribution using D’Augostino-Pearson omnibus normality test or Shapiro–Wilk normality test (samples sizes too small for D’Augostino–Pearson omnibus normality test). Afterwards all parameters were analyzed using a one-way ANOVA followed by Tukey post hoc test or a Kruskal–Wallis test followed by Dunn’s post hoc test, depending on Gaussian distribution. Data are presented as median with min to max or mean ± SEM depending on statistical analysis. Significance was taken at *p* ≤ 0.05.

## 3. Results

### 3.1. NVP-13 CNS, Liver, and Kidney Concentrations

Since these experiments are designed as GLP-toxicity studies in healthy animals and not in disease model animals, it was not in scope to assess the individual animal’s TGFβ signaling status. To assess NVP-13 tissue concentrations following 13 consecutive weeks of NVP-13 administration, as well as another 13 weeks of recovery period, NVP-13 tissue concentrations were measured within CNS, liver, and kidney samples (Figure 2).

### 3.2. Effects of NVP-13 on Physiological Parameters

To assess the effects of repeated NVP-13 administrations on physiological parameters, body weight development (Figure 3), body temperature (Figure 3), and food consumption (data not shown) were monitored throughout the entire experiment (for details see study plan). For none of the three abovementioned parameters were any unusual changes noted.

### 3.3. Effects of NVP-13 on Neurobehavioral Parameters

To assess effects of repeated NVP-13 administrations on neurobehavioral parameters, a standard observation battery, which allowed the assessment of peripheral and central nervous system activities, using a *modified* version of primary observation test described by Irwin for detecting neurological and behavioral tests (Irwin, 1968), was performed.

Here, no adverse or test item-related findings were noted (Appendix A).

### 3.4. Effects of NVP-13 on Physical Parameters

To assess the effects of repeated NVP-13 administrations on physical parameters, physical examinations were performed on all unsedated animals once during the predose phase, 1 h prior to dosing, and 4 h postdose on days 1, 15, 29, 57, and 85 and at the end of the recovery phase. Here, no adverse or test item-related findings were noted.

### 3.5. Effects of NVP-13 on Neurological Findings—Spinal Reflexes

Treatment with NVP-13 resulted in a transient absence of food and patellar reflexes (patellar mainly); these reflexes were present again after a maximum of 48 h. The transient absence of reflexes occurred in control animals as well but had a non-significant higher incidence in animals administered 2.0 or 4.0 mg/animal. No other neurological abnormalities were observed.

### 3.6. Effects of NVP-13 on Ophthalmic Parameters

To assess the effects of repeated NVP-13 administrations on ophthalmic parameters, animals were anesthetized with ketamine and medotomidine, and a mydriatic agent (1% tropicamide) was injected into the eyes prior to examination. Atipamezole was used as an antidote at the end of the investigation. All investigations were performed on all animals once during the predose phase, during week 13 of the dosing phase, and on all surviving animals during the last week of recovery phase. Macroscopic examinations were performed on the eye and connected tissues. The ocular fundus, with macula lutea, papilla, ocular vessels, and retina, was examined. Anterior and medium segments, with conjunctiva, cornea, anterior chamber, iris, lens, and vitreous body, were examined by slit-lamp microscopy. Fluorescein (0.5% in purified water) was applied as eye drops for epithelial staining. For none of these parameters were adverse or test item-related findings noted.

### 3.7. Effects of NVP-13 on Cardiovascular Parameters

To assess the effects of repeated NVP-13 administrations upon the cardiovascular system, the following investigations were performed in all animals (non-anesthetized, temporarily restrained animals), once during the predose phase, during weeks 4 and 13 of the dosing phase, and for all surviving animals during the last week of the recovery phase:-Electrocardiography: An eight-lead ECG measurement (Leads I, II, III, aVR, aVL, V1, and V2) was performed, and the following parameters were measured: heart rate (beats/min), RR, PR, QRS, QT, corrected QT.-Blood Pressure: Systolic, diastolic, and mean arterial pressures (mmHg) were recorded in all animals by high definition oscillometry (HDO) method.-Respiratory Rate: Investigations were performed on non-anesthetized, temporarily restrained animals once during the predose phase, in weeks 4 and 13 of the dosing phase, and during the last week of recovery phase by counting respiratory phases for 15 s for calculation of respiratory frequency (respirations/minute). For none of the measured parameters were any adverse or test item-related changes noted (Figure 3).

### 3.8. Effects of NVP-13 on Hematological Parameters

To assess the effects of repeated NVP-13 administrations on hematological parameters, blood samples were withdrawn from all animals once during the predose phase, during week 13 of the dosing phase, and from all surviving animals during the last week of the recovery phase, and the parameters given in Appendix A were analyzed. For none of the measured parameters were any adverse or test item-related changes noted (Figure 4).

### 3.9. Effects of NVP-13 on Coagulation

To assess effects of repeated NVP-13 administrations on coagulation, prothombin time (PT.) and activated partial thromboplastin time (APTT) were determined from blood collected into trisodium citrate anticoagulant. For neither of the two parameters were any adverse or test item-related changes noted (Figure 4).

### 3.10. Effects of NVP-13 on Clinical Chemistry

To assess the effects of repeated NVP-13 administrations on clinical chemistry, the parameters given in Appendix A were analyzed. For none of the measured parameters were any adverse or test item-related changes noted (Figure 5).

### 3.11. Effects of NVP-13 on Cerebrospinal Fluid Clinical Chemistry and Cell Count

To assess effects of repeated NVP-13 administrations on cerebrospinal fluid, clinical chemistry, and cell count (parameters given in Appendix A), CSF samples were withdrawn from all animals during the predose phase, in week 13 of the dosing phase, and from all surviving animals during the last week of recovery phase. Samples were collected by lumbar puncture (0.5 mL in total). The animals were anesthetized with ketamine and medetomidine prior to examinations. Atipamezole was used as an antidote at the end of the investigations. A pencil-point needle for pediatric use was used. The exact site of sampling (between Level L2–L6) was stated in the raw data. Withdrawn volume was substituted with aCSF in all case. For none of the measured parameters except total protein and micro albumin concentrations were any significant adverse or test item-related changes noted (Figure 6).

### 3.12. Effects of NVP-13 on Urine Physiology

To assess the effects of repeated NVP-13 administrations on urine physiology, urine samples (after approximately 2 h without food) were collected from all animals once during the predose phase, during week 13 of the dosing phase, and from all surviving animals during the last week of the recovery phase. The parameters to be analyzed are given in Appendix A. For none of the measured parameters were any adverse or test item-related changes noted (Figure 7).

### 3.13. Effects of NVP-13 on Macroscopic Observations and Organ Weights

The few macroscopic observations were consistent with the expected spectrum of background pathology in cynomolgus monkeys of this origin and age, and no evidence suggested an effect of the test item. For none of the measured organ weights were any adverse or test item-related changes noted.

### 3.14. Effects of NVP-13 on Microscopic Observations

Test item-related microscopic changes were observed in the central nervous system (including the IT injection site) and the iliac lymph node. These changes occurred at all dose levels, with no dose relationship.

#### 3.14.1. Central Nervous System

Mononuclear cell infiltration was observed at all levels of the spinal cord; in the spinal nerve roots; and, to a lesser extent, in the brain at all dose levels, with no relationship to dose. Only one control had this finding, with minimal intensity at the lumbar spinal cord level. The incidence and intensity of this finding were highest at the level of the lumbar spinal cord (injection site and lumbar spinal cord samples). The infiltrates were perivascular and mostly in the meninges but were occasionally also in the neuropil for distal spinal cord samples (cauda equina). Minimal to moderate fibrosis affecting the meninges was observed in the lumbar spinal cord (including the IT injection site). A similar minimal focal change was also present in one control female. Given the incidence and intensity, this change was considered test item related, although possibly indirectly due to enhancement of the local inflammatory reaction following the i.th. injection. Gliosis was occasionally observed in the lumbar spinal cord/i.th. injection site. In one male administered 1.0 mg/animal and one control male, slight demyelination was also present. These changes were generally focal. Given their low incidence and intensity, localization, and distribution, these changes were considered related to the dosing procedure and not to the test item. Minimal neuronolysis in the dorsal root ganglia (which was recorded under the spinal nerve root) was observed in a few test-item treated animals. However, this is known to occur spontaneously and was also observed in one recovery control animal.

#### 3.14.2. Iliac Lymph Node Changes

Granular macrophages were observed in the iliac lymph nodes of one male administered 1.0 mg/animal (slight), one male administered 4.0 mg/animal (minimal), and two females administered 4.0 mg/animal (minimal and slight). The presence of granular macrophages in lymph nodes is frequent for monkeys administered ASOs. All other changes were background changes with no relationship to the test item (Table 1).

#### 3.14.3. Necropsy of Recovery Group

The same pattern of changes in the central nervous system occurred in animals sacrificed at the end of the recovery phase. However, the incidence and/or intensity were generally lower, which indicated ongoing recovery. Granular macrophages were present in the mesenteric lymph node of one animal administered 4.0 mg/animal (slight severity), but not in the iliac lymph nodes.

## 4. Discussion

The present 13-week GLP-tox study successfully challenged the novel LNA antisense oligonucleotide NVP-13, targeting TGFBR2 mRNA, to be a safe, non-toxic, and highly promising drug candidate to treat neurodegenerative disorders following repeated intrathecal administrations. The in vivo data are in line with the in vitro results we described earlier in more detail [39] and therefore confirm the strategy of targeting this central key pathogenic factor, a candidate for a broad variety of different CNS and non-CNS disorders. In vivo NVP-13 treatment for 13 consecutive weeks (Figure 1) led to dose-dependent ASO concentrations within spinal cord and brain tissue (Figure 2). Further, the NVP-13 tissue levels decreased with increasing distance of the respective area to the injection site, the lumbar part of the spinal cord. These results are in line with studies in mice, juvenile cynomolgus monkeys, human infants [41,42], and own in vivo data (Peters et al. in preparation). The distribution of ASOs throughout the CNS is likely achieved by intrinsic CSF dynamics including CSF turn over and constant movement, driven by CSF production, pulsatility, perfusion, and respiration. Further, CSF is exchanged persistently, which is facilitated by convective influx of CSF along the perivenous and periarterial space into the brain parenchyma through the glymphatic system. Thereby, the movement of macromolecules throughout the parenchyma is enabled. The interaction of these CSF-affecting dynamics is likely a key factor in the broad distribution of ASOs within the CNS [22]. NVP-13 levels were also detectable in peripheral organs, namely the kidney and the liver, at treatment-active drug levels, but also indicating the regular physiological pattern of substance distribution and decay. Following a 13-week recovery phase with no further dosing procedures, there were almost no measurable NVP-13 concentrations detectable within the different types of tissue (Figure 2).

Our data indicate that NVP-13 did not exert any effects on the metabolism of male and female cynomolgus monkeys, since there were no alterations in bodyweight throughout the entire experimental paradigm (Figure 3A). The absence of any physiological alterations including body temperature (Figure 3B), respiratory rate (Figure 3C), and cardiovascular functions (Figure 3D–L) fulfill the requirements to explore/exclude undesirable pharmacological effects on vital organ systems as outlined in ICH guideline S7A. This is in line with the safety pharmacology profile of ASOs for different molecular targets and organs and different routes of administration [22,23]. Antisense Oligonucleotides are well known for their pro-inflammatory activity [43]. Hematological parameters (Figure 4A–Q) including mean corpuscular volume, hemoglobin concentrations, hematocrit, prothrombin time, and blood cell counts are well described as readout parameters for the activation state of the immune system and inflammatory processes (pro- as well as anti-inflammatory), dietary deficiencies, and a broad variety of systemic diseases. The results of the present study in combination with the mild pro-inflammatory response following application of therapeutic relevant doses with ASOs in general [43] and NVP-13 in particular [40] strengthen the hypothesis for this intervention to be safe for patients. To increase the understanding of effects on the immune system, future clinical trials should include a comprehensive monitoring of the accompanying inflammatory responses in patients under real life conditions. No alterations in clinical chemistry parameters (Figure 5) including total bilirubin, blood urea, enzymes, dietary parameters, and different ions are in line with the safe and non-adverse effects of NVP-13 described for the present study so far. The results give strong evidence for no interference with metabolic and catabolic pathways, not affecting liver, kidney, or bile functionality. These results were strengthened by unaltered urine parameters (Figure 7A–H).

Intrathecal injection of NVP-13 is accompanied with a transient (reversible) reduction in or absence of the patellar reflex and food grip reflex. This was observed post-dose on all dosing days and occurred in all dose groups, with an increased incidence in animals administered 2.0 or 4.0 mg/animal. The severity and incidence were constant throughout the dosing phase. Changes in lower spinal reflexes were most often recorded 4 and 24 h post dose and were fully reversible within 48 h after dosing. These transient changes in spinal reflexes were considered non-adverse and represented an oligonucleotide class-specific finding after i.th. administration [44]. A recently published paper by Korte et al. described these effects as not necessarily ASO-related, since they also occurred in aCSF-treated animals, indicating the dosing procedure per se influences neurological parameters [45]. Further, persistent loss of one or more reflexes was described as non-adverse, since no fundamental physiological deficit or behavioral alterations were noted [44]. Following 13 weeks of intrathecal NVP-13 treatment, mononuclear cell infiltrates in the nervous system were considered to have resulted from proinflammatory effects of ASOs. Likewise, the slight meningeal fibrosis observed in the lumbar spinal cord (including intrathecal injection sites) of test item-treated animals likely represented enhancement of the local inflammatory response following the dosing procedure [43].

In addition, the presence of granular macrophages in lymph nodes is known to be frequent in monkeys following i.th. ASO administrations. This finding was considered to represent the uptake of ASOs into lysosomes, an adaptive change that does not represent an adverse effect on cell structure or function [46]. Due to the absence of correlating clinical findings, these changes were considered non-adverse. Perivascular mononuclear cell infiltrates at the injection site were considered to have resulted from the well-known pro-inflammatory properties of antisense oligonucleotides [47] and were considered non-adverse given their low severity. Intrathecal administration of 1.0, 2.0, or 4.0 mg/animal NVP-13 once every 2 weeks for 3 weeks (seven injections) resulted in test-item related findings in the nervous system (spinal cord, including injection sites, and, to a lesser extent, brain) that consisted of perivascular mononuclear cell infiltrates and slight fibrosis, which were mainly located in the meninges. Findings were observed at all dose levels, with no relationship to dose. These findings were considered to be pro-inflammatory class effects of the test item. Accumulation of granular macrophages in the iliac lymph nodes was also noted in a few animals administered 1.0 or 4.0 mg/animal, which represented accumulation of the test item in macrophages. In the absence of associated clinical observations, these changes were considered non adverse. At the end of the recovery phase, a similar pattern was observed, with a lower incidence and/or severity, which demonstrated ongoing recovery.

## 5. Conclusions

In conclusion, intrathecal administration of 1, 2, or 4 mg NVP-13/animal for 3 months on days 1, 15, 29, 43, 57, 71, and 85 in the initial 13 weeks showed good local and systemic tolerability and no adverse microscopic findings in male or female cynomolgus monkeys, matching the literature [46]. Under the conditions of this study, the no observed adverse effect level (NOAEL) was documented up to the highest dose level of 4 mg/animal NVP-13. These results, in conjunction with the recently described in vivo safety and efficacy of TGFBR2 mRNA downregulation, as well as the efficient reconditioning of the neurogenic niche of adult non-human primates, highlights NVP-13 as a potential and promising drug candidate to treat central key pathogenic factors relevant for many disorders, as mentioned above. In addition, based on the outcome of the core activities of modifying TGF-β signaling, namely reverting stem cell exhaustion, modifying the immune response, and having a strong anti-fibrotic (extracellular matrix rejuvenation) and proautophagic action [1,2,33,38,39], this drug candidate is worthwhile for further differentiated evaluation of its potential in rejuvenation/longevity strategies.

## Figures and Tables

**Figure 1 pharmaceutics-14-00200-f001:**
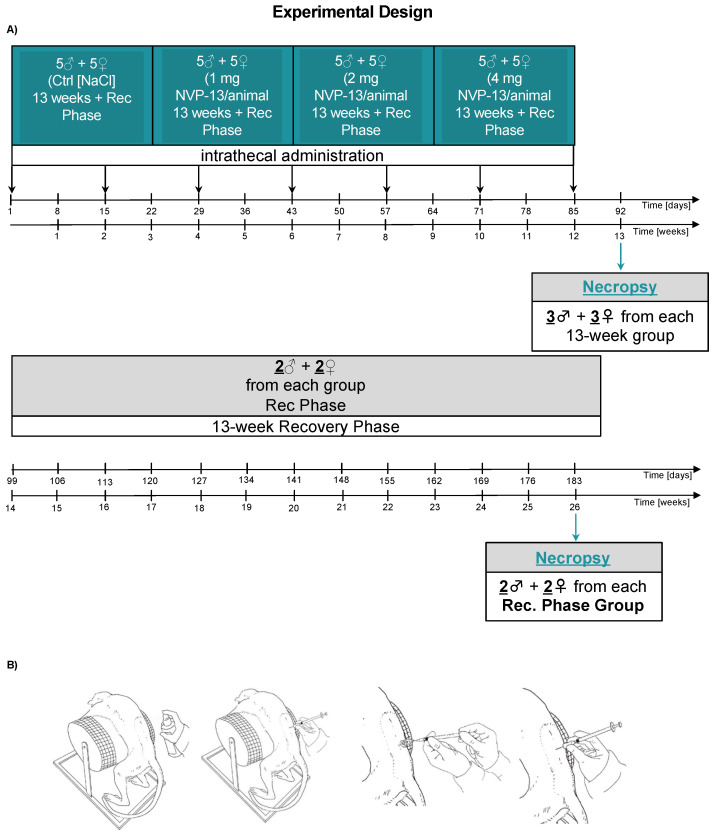
Schematic illustration of the experimental design (**A**) and the dosing procedure (**B**). First, 0.9% NaCl (saline control, n = 6) and NVP-13 (1 mg, n = 6; 2 mg, n = 6; 4 mg, n = 6) were injected repeatedly over a 13-week period with a subsequent 13 week recovery phase (**A**). Following disinfection, intrathecal (i.th.) administration was performed via lumbar puncture between L3-L5 by slow manual bolus infusion over 1 min to anesthetized animals. The needle (with syringe) was left in the dosing site for at least 30 s after an aCSF flush. It was documented that CSF flow was present before dosing, that the position of the needle opening was facing towards the head of the animal prior to dose administration, and that the needle (with syringe) was left in dosing site for at least 30 s after aCSF flush (**B**).

**Figure 2 pharmaceutics-14-00200-f002:**
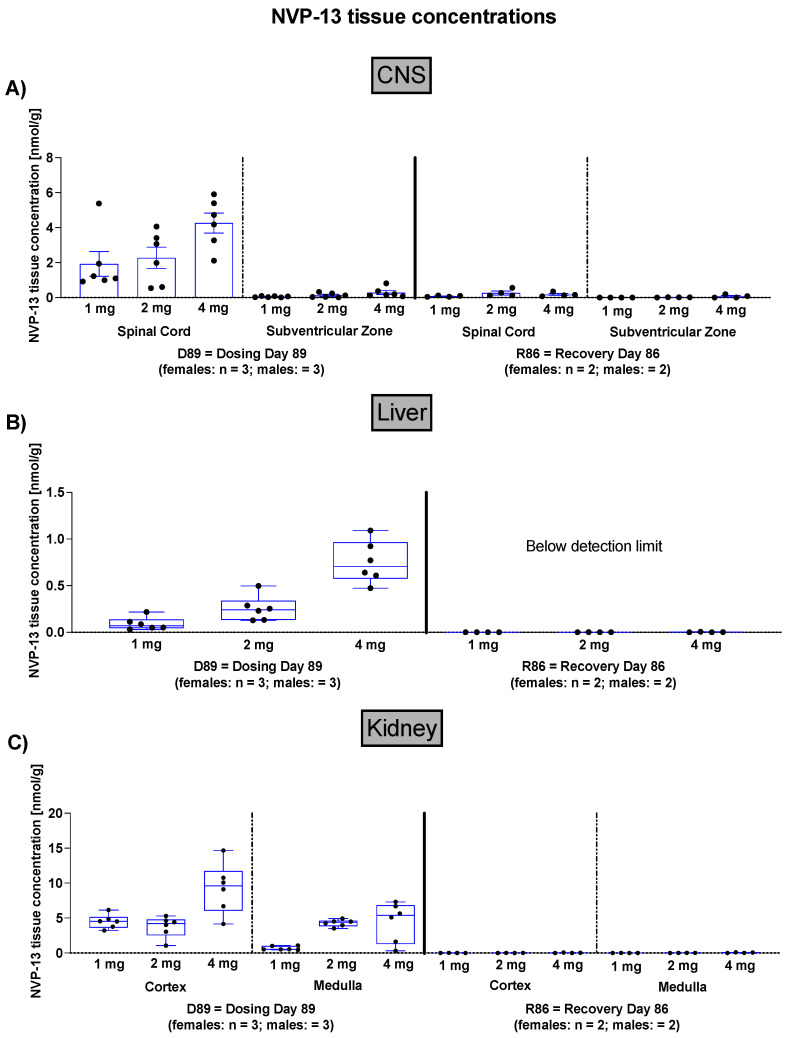
NVP-13 CNS (**A**), liver (**B**), and kidney (**C**) concentrations. NVP-13 treatment for 13 consecutive weeks resulted in detectable tissue concentrations within the CNS, the liver, and the kidney that remained measurable after a 13-week recovery period without any treatment within the CNS, but not the liver and kidney. All parameters were tested for Gaussian distribution using D’Augostino-Pearson omnibus normality test or Shapiro–Wilk normality test (samples sizes too small for D’Augostino–Pearson omnibus normality test). Afterwards, all parameters were analyzed using a one-way ANOVA followed by Tukey post hoc test or a Kruskal–Wallis test followed by Dunn’s post hoc test, depending on Gaussian distribution. Data are presented as median with min to max or mean ± SEM depending on statistical analysis. Significance was taken at *p* ≤ 0.05.

**Figure 3 pharmaceutics-14-00200-f003:**
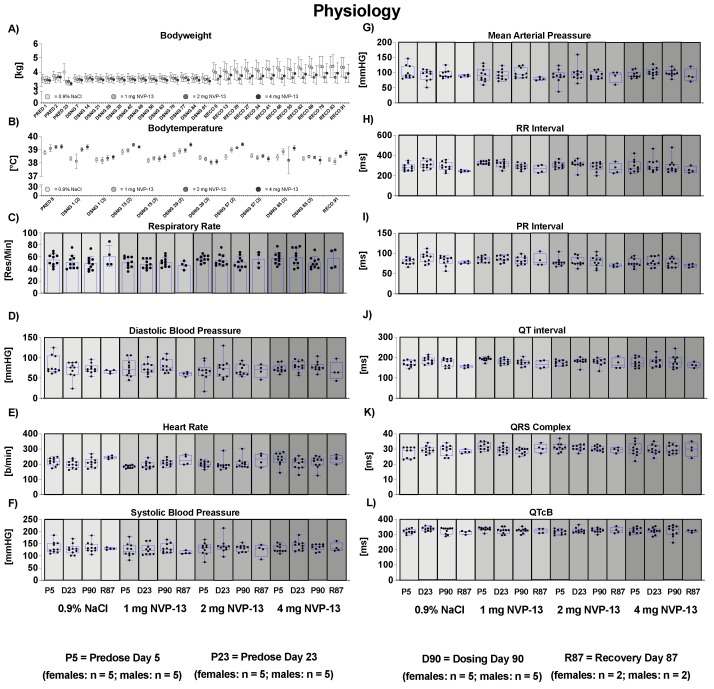
Effects of NVP-13 on body weight development (**A**), body temperature (**B**), respiratory rate (**C**), and cardiovascular parameters (**D**–**L**). NVP-13 treatment for 13 consecutive weeks had no effects on body weight development, body temperature, respiratory rat, and cardiovascular parameters. All parameters were tested for Gaussian distribution using D’Augostino–Pearson omnibus normality test or Shapiro–Wilk normality test (samples sizes too small for D’Augostino–Pearson omnibus normality test). Afterwards, all parameters were analyzed using a one-way ANOVA followed by Tukey post hoc test or a Kruskal–Wallis test followed by Dunn’s post hoc test, depending on Gaussian distribution. Data are presented as median with min to max or mean ± SEM depending on statistical analysis. Significance was taken at *p* ≤ 0.05.

**Figure 4 pharmaceutics-14-00200-f004:**
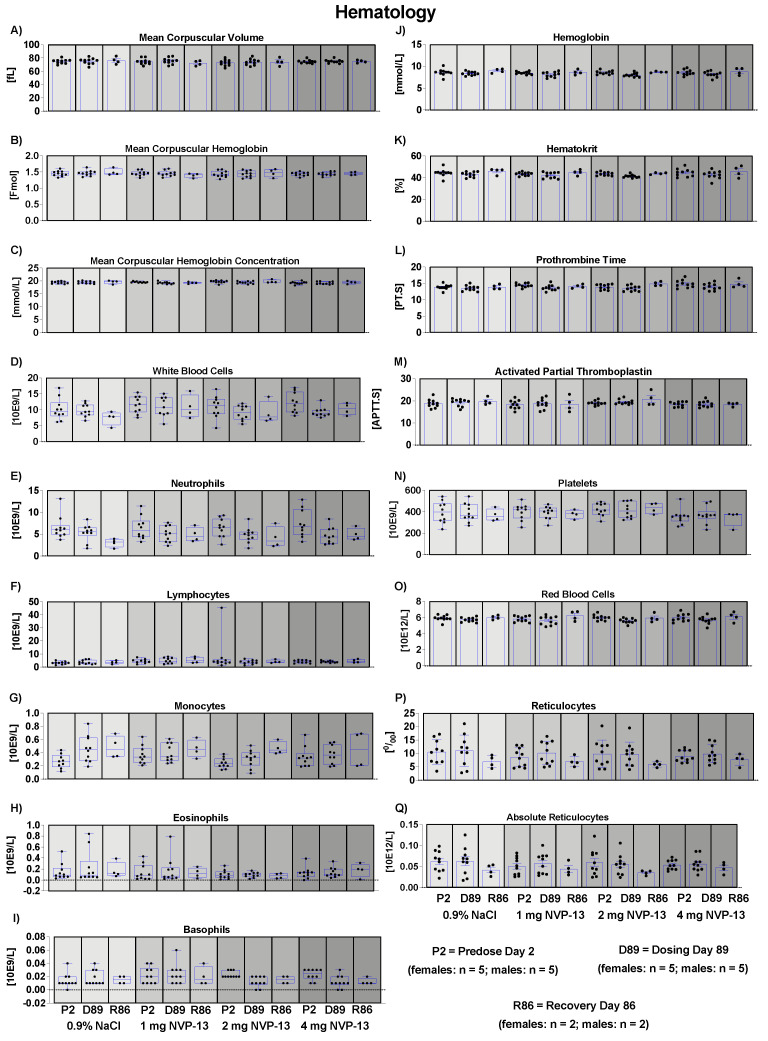
Effects of NVP-13 on hematological parameters (**A**–**Q**): NVP-13 treatment for 13 consecutive weeks had no effects on hematological parameters. All parameters were tested for Gaussian distribution using D’Augostino-Pearson omnibus normality test or Shapiro-Wilk normality test (samples sizes too small for D’Augostino-Pearson omnibus normality test). Afterwards, all parameters were analyzed using a one-way ANOVA followed by Tukey post hoc test or a Kruskal-Wallis test followed by Dunn’s post hoc test, depending on Gaussian distribution. Data are presented as median with min to max or mean ± SEM depending on statistical analysis. Significance was taken at *p* ≤ 0.05.

**Figure 5 pharmaceutics-14-00200-f005:**
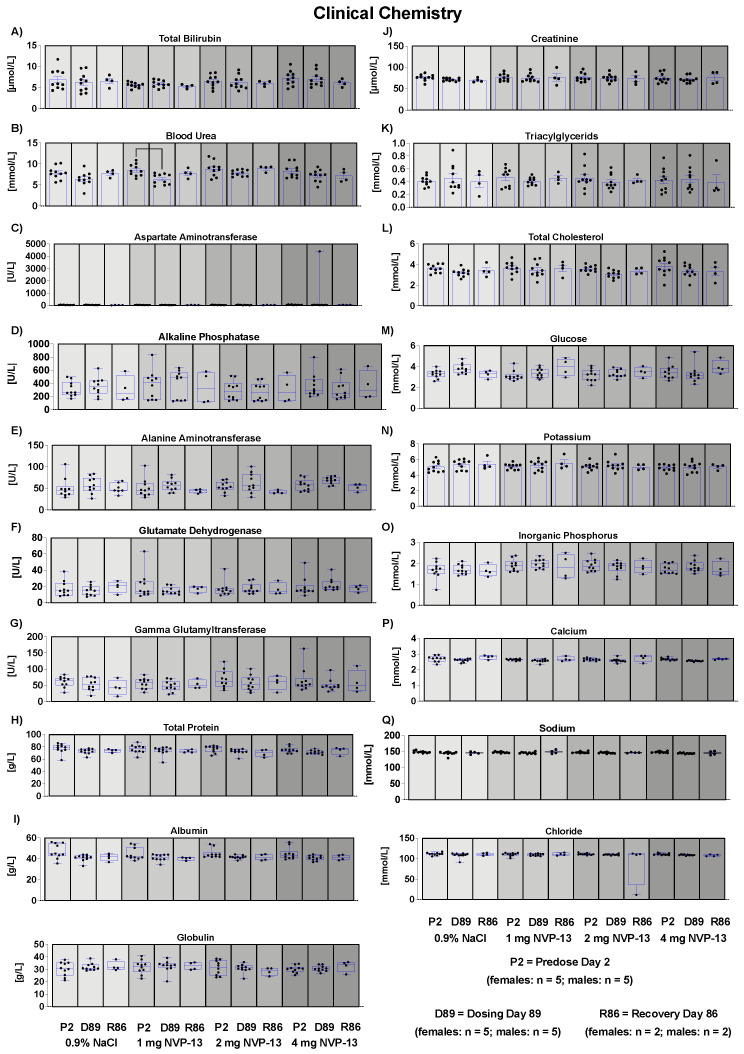
Effects of NVP-13 on clinical chemistry parameters (**A**–**Q**): NVP-13 treatment for 13 consecutive weeks had no effects on clinical chemistry. All parameters were tested for Gaussian distribution using D’Augostino-Pearson omnibus normality test or Shapiro-Wilk normality test (samples sizes too small for D’Augostino-Pearson omnibus normality test). Afterwards, all parameters were analyzed using a one-way ANOVA followed by Tukey post hoc test or a Kruskal-Wallis test followed by Dunn’s post hoc test, depending on Gaussian distribution. Data are presented as median with min to max or mean ± SEM depending on statistical analysis. Significance was taken at *p* ≤ 0.05.

**Figure 6 pharmaceutics-14-00200-f006:**
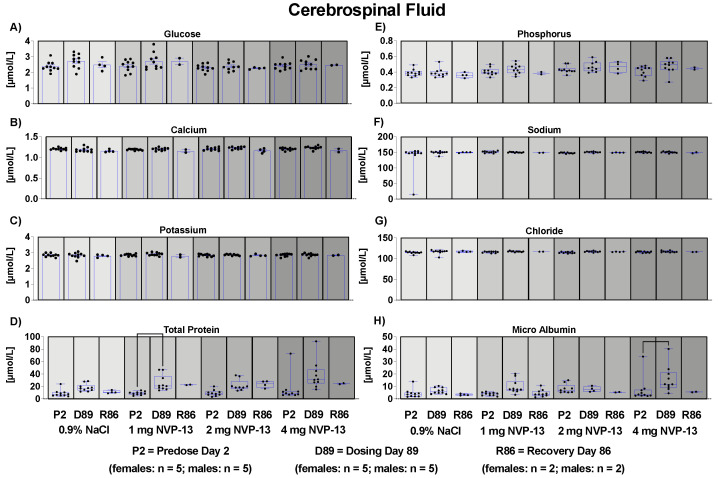
Effects of NVP-13 on CSF biochemistry parameters (**A**–**H**): NVP-13 treatment for 13 consecutive weeks had almost no effects on CSF parameters. Total protein and micro albumin concentrations were enhanced on D89 following 13 weeks of 1mg NVP-13 and 4 mg NVP-13, respectively (**D**,**H**). All parameters were tested for Gaussian distribution using D’Augostino-Pearson omnibus normality test or Shapiro-Wilk normality test (samples sizes too small for D’Augostino-Pearson omnibus normality test). Afterwards, all parameters were analyzed using a one-way ANOVA followed by Tukey post hoc test or a Kruskal-Wallis test followed by Dunn’s post hoc test, depending on Gaussian distribution. Data are presented as median with min to max or mean ± SEM depending on statistical analysis. Significance was taken at *p* ≤ 0.05.

**Figure 7 pharmaceutics-14-00200-f007:**
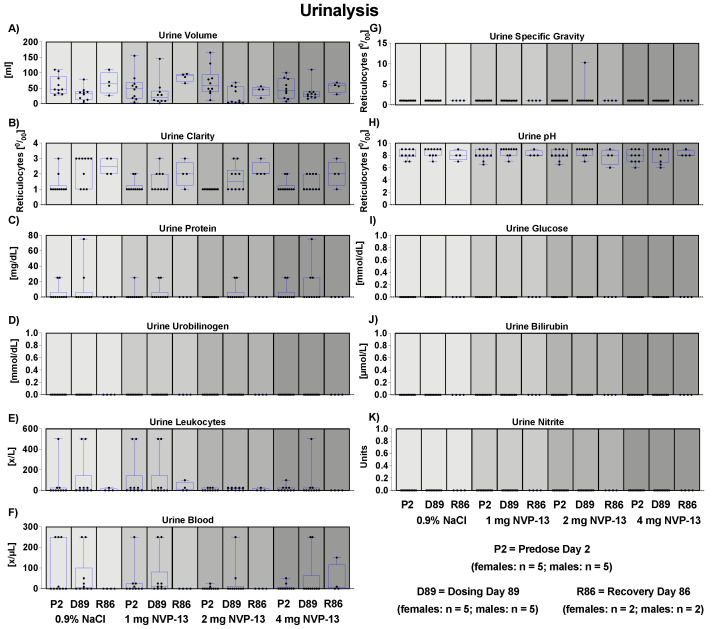
Effects of NVP-13 on different urine parameters (**A**–**K**). NVP-13 treatment for 13 consecutive weeks had no effects on urine parameters. All parameters were tested for Gaussian distribution using D’Augostino-Pearson omnibus normality test or Shapiro-Wilk normality test (samples sizes too small for D’Augostino-Pearson omnibus normality test). Afterwards, all parameters were analyzed using a one-way ANOVA followed by Tukey post hoc test or a Kruskal-Wallis test followed by Dunn’s post hoc test, depending on Gaussian distribution. Data are presented as median with min to max or mean ± SEM depending on statistical analysis. Significance was taken at *p* ≤ 0.05.

**Table 1 pharmaceutics-14-00200-t001:** Effects of NVP-13 on microscopic observations. NVP-13 administration for 13 consecutive days resulted in minimal to moderate microscopic histological alterations that were dose unspecific and present in any group.

Sex	NVP-13							
	Males				Females			
Dose Level (mg/Animal)	0	1	2	4	0	1	2	4
**Brain**								
Number Examined	2	2	2	2	2	2	2	2
Infiltrate, mononuclear cells								
Not Present	1	1	2	1	2	1	1	1
Minimal	1	1	0	1	0	1	1	1
**Ganglion, Trigeminal**								
Number Examined	2	2	2	2	2	2	2	2
Infiltrate, mononuclear cells								
Not Present	2	2	2	2	2	2	2	1
Minimal	0	0	0	0	0	0	0	1
**Intrathecal, Injection site**								
Number Examined	2	2	2	2	2	2	2	2
Fibrosis								
Not Present	2	1	2	1	2	2	2	1
Minimal	0	0	0	0	0	0	0	1
Slight	0	1	0	1	0	0	0	0
**Intrathecal, Injection site**								
Number Examined	2	2	2	2	2	2	2	2
Gliosis, NOS								
Not Present	2	1	2	2	2	2	2	2
Minimal	0	1	0	0	0	0	0	0
**Intrathecal, Injection site**								
Number Examined	2	2	2	2	2	2	2	2
Infiltrate, mononuclear cells								
Not Present	1	1	1	1	2	1	1	1
Minimal	1	1	1	1	0	1	1	1
**Spinal cord, Cervical**								
Number Examined	2	2	2	2	2	2	2	2
Infiltrate, mononuclear cells								
Not Present	2	1	2	2	2	2	1	2
Minimal	0	1	0	0	0	0	1	0
**Spinal cord, Lumbar**								
Number Examined	2	2	2	2	2	2	2	2
Fibrosis, focal								
Not Present	2	1	2	1	2	2	2	0
Minimal	0	0	0	1	0	0	0	1
Slight	0	1	0	0	0	0	0	1
**Spinal cord, Lumbar**								
Number Examined	2	2	2	2	2	2	2	2
Gliosis, NOS								
Not Present	2	1	2	2	2	2	0	1
Minimal	0	1	0	0	0	0	2	1
**Spinal cord, Lumbar**								
Number Examined	2	2	2	2	2	2	2	2
Infiltrate, mononuclear cells								
Not Present	2	1	1	0	2	0	1	1
Minimal	0	1	1	2	0	2	0	0
Slight	0	0	0	0	0	0	1	0
Moderate	0	0	0	0	0	0	0	1
**Spinal cord, Thoracic**								
Number Examined	2	2	2	2	2	2	2	2
Infiltrate, mononuclear cells								
Not Present	2	1	2	1	2	1	2	2
Minimal	0	1	0	1	0	1	0	0
**Spinal nerve roots**								
Number Examined	2	2	2	2	2	2	2	2
Infiltrate, mononuclear cells								
Not Present	2	1	1	2	2	2	1	2
Minimal	0	1	0	0	0	0	0	0
Slight	0	0	1	0	0	0	1	0
**Spinal nerve roots**								
Number Examined	2	2	2	2	2	2	2	2
Neuronolysis, dorsal root ganglia								
Not Present	2	2	2	2	1	2	1	2
Minimal	0	0	0	0	1	0	1	0
**Lymph node, Mesenteric**								
Number Examined	2	2	2	2	2	2	2	2
Granular macrophages								
Not Present	2	2	2	1	2	2	2	2
Slight	0	0	0	1	0	0	0	

## Data Availability

The authors declare that [the/all other] data supporting the findings of this study are available within the paper [and its Appendix A]. Additional information/data on NVP-13 that support the findings of this study are available from the corresponding author upon reasonable request.

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
