# Peer review of "Safe and Effective Cynomolgus Monkey GLP—Tox Study with Repetitive Intrathecal Application of a TGFBR2 Targeting LNA-Gapmer Antisense Oligonucleotide as Treatment Candidate for Neurodegenerative Disorders"

_pharmaceutics, 2022, doi:10.3390/pharmaceutics14010200_

Round 1

Reviewer 1 Report

The paper by Peters et al. presents a GLP-Tox study in cynomolgus monkeys of a lead therapeutic candidate, namely, an LNA gapmer antisense oligonucleotide (ASO) NVP-13 for TGFBR2 downregulation as a potential treatment for amyotrophic lateral sclerosis (ALS). The paper describes valuable data on tolerability and adverse effects of the compound, which would be instructive to many researchers interested in the application of ASOs to treat neurodegenerative diseases, and is certainly worth publishing after minor revision.

Major points

Introduction

An additional paragraph or two on the application of ASOs to treat neurodegenerative disorders such as Alzheimer, Parkinson or Huntington diseases would be a welcome addition to those readers who is not deeply familiar with the topic.

More specifically, other ASO-based approaches to the treatment of ALS need to be referenced and compared to the TGFBR2 downregulation approach adopted in the paper.

2.1 Antisense oligonucleotide (NVP-13) characteristics

Sequence and structure (pattern of chemical modifications) of NVP-13 need to be included.

Figures

I would expect to see more Hi-Res versions of the Figures and the Tables to appear in the revised version.

Conclusions

The last sentence of the conclusion (lines 544-546) seems to be somewhat farfetched. I would like to see more evidence in the Discussion section to support this point.

Minor points

The manuscript has quite a few typos, which require thorough checking. A (non-exhaustive) list of these picked up by this reviewer is given below.

Title, line 3

I suggest to avoid abbreviations in the title. Please replace “i.th.” with “intrathecal”. The correct term is “gapmer”, not “gapmere”.

Abstract

Lines 29-30: antisense oligonucleotide NP13.

Lines 35-37: “We were able to demonstrate intrathecal administration of 1, 2, or 4 mg NVP-13/animal within 3 months on days 1, 15, 29, 43, 57, 71, and 85 in the initial 13 weeks.” The meaning of the sentence is unclear. Does it mean: “NVP-13 was administered intrathecally at 1, 2, or 4 mg per animal on days 1, 15, 29, 43, 57, 71, and 85 of the initial 13 weeks”?

Introduction

Line 82: add full stop.

Line 84: delete full stop. Same line: “We recently showed that…”

Line 90: β is missing.

Line 92: “LNA-gapmer”.

Line 93: “We previously showed that treatment with NVP-13 targeting the TGFBR2 mRNA reduced TGFBR2 expression…”

Materials and methods

Lines 112-113: references are needed for in vitro and SMA data mentioned.

Figure 2, A): the X axis has two sets of labels.

Discussion

Line 481: “(pro- as well as anti-inflammatory…”

Reviewer 2 Report

The Peters et al. manuscript describes a thorough safety investigation of NVP13, a LNA ASO that targets TGFBR2, considered to be a treatment option of neurodegenerative disorders.

Figure 2: The labels in graph A overlap making data interpretation difficult. 

This figure shows NVP-13 levels in tissues of interest – what were the TGFBR2 expression levels and TGFβ signalling levels in these tissues? Specifically, the CNS tissues where the drug is hoped to have therapeutic effects for neurodegenerative disorders. Confirming the ASO was having the predicted effect in the model animals used for the safety and efficacy tests would help to boost the confidence in the results shown.  

In figures 3, 4, 5, 6 and 7 the sample labels on the X axis of the graphs are inconsistent, redundant and/or overlapping. Please make consistent and clear. I would also place the key at the top of the figure not at the bottom for ease of understanding.

In figure 3 – graphs A and B the data are represented by different symbols (dot and circle) – what do these different symbols represent?

In figure 6 graphs D and H there are what look like significance lines comparing samples, however the p values of these samples are not mentioned in the figure or figure legend nor are these discussed in the text.

There are some font and nomenclature inconsistencies in the document.

Round 2

Reviewer 2 Report

Revised manuscript fine for publication.